# Selective Anticancer Therapy Based on a HA-CD44 Interaction Inhibitor Loaded on Polymeric Nanoparticles

**DOI:** 10.3390/pharmaceutics14040788

**Published:** 2022-04-04

**Authors:** José M. Espejo-Román, Belén Rubio-Ruiz, Victoria Cano-Cortés, Olga Cruz-López, Saúl Gonzalez-Resines, Carmen Domene, Ana Conejo-García, Rosario M. Sánchez-Martín

**Affiliations:** 1Department of Medicinal and Organic Chemistry and Excellence Research Unit of Chemistry Applied to Biomedicine and the Environment, Faculty of Pharmacy, Campus Cartuja s/n, University of Granada, 18071 Granada, Spain; jmespejo@ugr.es (J.M.E.-R.); belenrubio@ugr.es (B.R.-R.); vccortes@go.ugr.es (V.C.-C.); olgacl@ugr.es (O.C.-L.); 2GENYO, Centre for Genomics and Oncological Research, Pfizer/University of Granada/Andalusian Regional Government, PTS Granada, Avda. Ilustración 114, 18016 Granada, Spain; 3Biosanitary Institute of Granada (ibs.GRANADA), SAS-University of Granada, Avenida de Madrid, 15, 18012 Granada, Spain; 4Department of Chemistry, University of Bath, Claverton Down, Bath BA2 7AX, UK; s.gonzalez.resines@bath.ac.uk (S.G.-R.); mcdn20@bath.ac.uk (C.D.); 5Chemistry Research Laboratory, University of Oxford, Mansfield Road, Oxford OX1 3TA, UK

**Keywords:** nanomedicine, selective release, anticancer therapy, hyaluronic acid, cluster of differentiation 44, tetrahydroisoquinoline, molecular dynamics simulations

## Abstract

Hyaluronic acid (HA), through its interactions with the cluster of differentiation 44 (CD44), acts as a potent modulator of the tumor microenvironment, creating a wide range of extracellular stimuli for tumor growth, angiogenesis, invasion, and metastasis. An innovative antitumor treatment strategy based on the development of a nanodevice for selective release of an inhibitor of the HA-CD44 interaction is presented. Computational analysis was performed to evaluate the interaction of the designed tetrahydroisoquinoline-ketone derivative (**JE22**) with CD44 binding site. Cell viability, efficiency, and selectivity of drug release under acidic conditions together with CD44 binding capacity, effect on cell migration, and apoptotic activity were successfully evaluated. Remarkably, the conjugation of this CD44 inhibitor to the nanodevice generated a reduction of the dosis required to achieve a significant therapeutic effect.

## 1. Introduction

Hyaluronic acid or hyaluronan (HA), the main component of the extracellular matrix, is a linear polysaccharide composed of repeating disaccharide units of *N*-acetyl-D-glucosamine and D-glucuronic acid with β-(1→4) interglycosidic linkages. In normal physiological conditions, the number of repeating disaccharides in an HA molecule ranges from 2000 to 25,000, resulting in a viscose and elastic solution with a large hydrodynamic volume that helps to maintain tissue integrity and homeostasis [1].

Besides its key role as structural component of tissues, HA is also involved in multiple signaling pathways, under both physiological (embryogenesis) and pathological conditions such as inflammation or cancer [2,3]. This unique biological function is attributed to its specific binding and interactions with HA-binding proteins, termed hyaladherins, which comprise several types of cell-surface receptors. Of these, cluster of differentiation 44 (CD44), a signal-transducing glycoprotein, is the major surface HA receptor, which is implicated in a variety of cellular events such as cell proliferation, differentiation, migration, and angiogenesis [4,5]. There is clear evidence that extensive HA production due to an aberrant synthesis or turnover occurs during malignant transformation. These abnormal HA levels are strongly associated with tumor aggressiveness and a fatal disease outcome. Likewise, the expression of CD44 is elevated in many types of malignancies compared to CD44 levels in the corresponding healthy tissues. Pathological conditions also promote alternate splicing and post-translational modifications, resulting in diversified CD44 molecules with enhanced HA binding leading to increased tumorigenicity [6,7]. Thus, HA acts as a potent modulator of tumor microenvironments through its interactions with CD44. Consequently, targeting the interactions between HA and CD44 is a promising approach against HA-induced tumorigenesis. 

Nanotechnology has made a remarkable contribution to cancer diagnosis and, importantly, cancer therapy [8,9]. In order to provide more effective and safer treatments, several nanodevices targeting CD44 receptors have been reported to specifically carry and deliver drugs [10,11,12,13]. The design of these nanosystems is based on the use of HA as the ligand for selective delivery of therapy on tumor cells overexpressing CD44. Moreover, a novel strategy has been recently reported using anti-CD44 antibody as the ligand for selective delivery of paclitaxel for treatment of pancreatic cancer [14]. However, a nanotechnology-based strategy for the inhibition of CD44 receptors as an anticancer therapeutic approach has not been reported so far. 

Crystal structure analysis together with mutagenesis studies of both murine and human CD44 have pinpointed the essential residues for HA binding to CD44 [15,16]. The CD44 HA-binding domain (CD44 HABD) is in the N-terminal domain at the extracellular region of the receptor. By means of biophysical binding assays, fragment screening, and crystallographic characterization of complexes with CD44 HABD, Liu LK et al. discovered an inducible pocket adjacent to the HA-binding groove in which small tetrahydroisoquinoline (THIQ)-containing molecules bind [16]. Among them, the **THIQ-ester** derivative (Figure 1) showed a significant affinity for the isolated protein. 

Based on our broad expertise in the functionalization of nanoparticles for the selective delivery of biomolecules [17,18,19,20,21,22], our aim was to implement a nanotechnology strategy to enhance the efficiency of the THIQ derivatives targeting CD44 to achieve a potential antitumor treatment. We first designed and synthesized an analogue of the reported CD44 inhibitor (**THIQ-ester**, Figure 1) [16] by replacement of the ester functional group by a ketone (**JE22**, Figure 1) to allow for the conjugation to the nanospheres via the hydrazone bond. This nanodevice **JE22-NPs** (**5**) selectively releases this THIQ derivative as a specific inhibitor of the HA–CD44 interaction at the acidic tumor microenvironment (Figure 1).

## 2. Materials and Methods

### 2.1. Computational

#### 2.1.1. Systems Set Up

The crystal structure of the murine CD44 HABD at 1.4 Å resolution (PDB ID: 5BZK) [16] was used as a starting point for the computational work. The protein was inserted in a water box of 90 × 90 × 90 A^3^ dimensions, and KCl was added up to a final concentration of 150 mM using the CHARMM-GUI Solution Builder server [23,24,25,26]. Two independent systems were built with the **THIQ-ester** and **JE22** ligands, respectively. The **JE22** molecule was aligned using the position of the **THIQ-ester** ligand present in PDB id 5BZK. The final systems were composed of ~66,000 atoms. Five different replicas were run for each system.

#### 2.1.2. Molecular Dynamics Simulations

The CHARMM36 force field [27,28] was used to model the protein, standard CHARMM parameters were used for ions, and the TIP3P model for water [29]. The charges and parameters for the ligands were searched using the CHARMM-GUI ligand modeler interface [30,31] that generates the ligand force field parameters and necessary structure files by inspecting small molecules in the verified CHARMM force field library or using the CHARMM general force field (CGenFF) [32]. The results suggested optimizing the charges and certain dihedral angles as the penalties were high. Geometry, charge, and dihedral optimization were performed for the two ligands following a standard protocol described in the Appendix A). The protocol for the validation and optimization of the few parameters with high penalties was the same as CGenFF using the FFTK plugin tool of VMD as an input generator and refinement tool. The target data were generated with several quantum ab initio methods in Gaussian16 [33]. The penalty score returned for every bonded parameter and charge was used to guide the selective optimization of the charges and some dihedral angles; bonds, angles, and improper force constants did not require any optimization. The same equilibration protocol was used for all the simulations (see Appendix A for details). The analysis was performed using in-house python scripts and the pyemma (http://emma-project.org/latest/, accessed on 5 March 2022) and mdtraj (https://www.mdtraj.org/, accessed on 5 March 2022) analysis tools.

### 2.2. Chemistry and Characterization

#### 2.2.1. General

All chemical reagents were supplied by Sigma-Aldrich. Gibco (Thermo Fisher Scientific, Waltham, MA, USA) was the supplier for the biological products including fetal bovine serum (FBS), trypsin-EDTA, Dulbecco’s modified Eagle’s medium (DMEM), 1% penicillin/streptomycin, and L-glutamine. Unlabeled HA (50 KDa) and HA-FITC (50 KDa) were purchased from HAworks LLC. Anti-CD44-FITC and CD44 monoclonal antibodies (MA5-15462, 8E2F3) were purchased from Miltenyi Biotec and Invitrogen, respectively. 

An Eppendorf Thermomixer^®^ agitator (Eppendorf, Hamburg, Germany) was used for conjugations, while centrifugations were performed in an Eppendorf centrifuge. Analytical TLC was performed using Merck Kieselgel 60 F_254_ aluminum plates and visualized by UV light. Evaporation was carried out in vacuo in a Büchi rotary evaporator, and the pressure controlled by a Vacuubrand CVCII apparatus. Purifications were carried out by flash column chromatography using silica gel (230–440 mesh ASTM, Merck KGaA, Darmstadt, Germany).

Nuclear magnetic resonance (NMR) spectra were recorded on a 400 MHz ^1^H and 101 MHz ^13^C NMR with a Varian Direct Drive spectrometer at room temperature. Chemical shifts (δ) are reported in parts per million (ppm) relative to the residual solvent peak. The multiplicity of each signal is given as s (singlet), d (doublet), t (triplet), and m (multiplet). *J* values are given in Hz. High-Resolution Electrospray Ionization (ESI-TOF) mass spectra were carried out on a Waters LCT Premier Mass Spectrometer. 

#### 2.2.2. Synthesis of 4-(3,4-Dihydroisoquinolin-2(1*H*)-yl)butan-2-one (**JE22**)

Compound **JE22** was synthesized as previously reported (Figure 1), and the spectroscopic data are in agreement with those reported in the literature [34]. See Appendix A for details.

#### 2.2.3. Preparation of Therapeutic Polymeric Nanoparticles **JE22-NPs** (**5**) 

The synthesis of the polymeric nanoparticles **JE22-NPs** (**5**) is displayed in Figure 1. Aminomethyl NPs (**Naked-NPs**, **1**) (1 mL, 3% SC, 64 μmol/g, 1 μmol, 1 eq.) were synthesized as previously reported (see Appendix A for synthetic details) [17]. Then, **Naked-NPs** (**1**) were conditioned in *N*,*N*-dimethylformamide (DMF) (1 mL × 3 times). N-α-Fmoc-glycine (Fmoc-Gly-OH) (50 eq.) was mixed in DMF (1 mL) with oxyme (50 eq.) and *N*,*N*′-diisopropylcarbodiimide (DIC) (50 eq.) for 10 min at 25 °C. Then, this solution was added to **Naked-NPs** (**1**) and stirred for 2 h at 60 °C at 1400 rpm on the Thermomixer. Subsequently, the NPs were washed by centrifugation (13,400 rpm, 3–10 min) with DMF, MeOH, and water to obtain Fmoc-Gly-NPs (100% yield, 0.064 mmol g^−1^ of amino groups). Then, Fmoc deprotection with 20% piperidine/DMF (3 × 20 min) was carried out. Separately, Fmoc-4,7,10-trioxa-1,13-tridecanediamine succinamic acid (Fmoc-PEG-COOH) spacer (50 eq.) was dissolved in DMF (1 mL), then oxyme (50 eq.) and DIC (50 eq.) were added and mixed for 10 min at 25 °C, and this last solution was mixed to NPs for 2 h at 60 °C. Subsequently, Fmoc deprotection was carried out to give **PEGylated-NPs** (**2**). Next, a solution of succinic anhydride (50 eq.) and *N*,*N*-diisopropylethylamine (DIPEA) (50 eq.) in DMF (1 mL) was added to NPs, sonicated, and mixed for 2 h at 60 °C. Next, **COOH-NPs** (**3**) were activated with oxyme (50 eq.) and DIC (50 eq.) for 4 h at 25 °C. NPs were centrifuged, and a solution of 55% *v*/*v* hydrazine hydrate (75 eq.) in DMF (1 mL) was added, and NPs were left stirring at 25 °C for 15 h. Subsequently, **hydrazine-NPs** (**4**) were washed and conditioned in MeOH. Finally, **JE22** (10 eq.) was dissolved in MeOH (1 mL) and added to NPs with a drop of trifluoroacetic acid (TFA), and the resulting mixture was stirred at 25 °C for 15 h on the Thermomixer at 1400 rpm. **JE22-NPs** (**5**) were afforded by centrifugation and subsequently washed with DMF (3 × 1 mL), MeOH (3 × 1 mL), and sterile ultrapure water (3 × 1 mL) (Figure 1 and Appendix A). 

#### 2.2.4. Characterization of **JE22-NPs** (**5**)

Particle size distribution and mean size were measured using dynamic light scattering (DLS) with biological grade water in a disposable cuvette. Zeta potential values were determined on a Zetasizer Nano ZS ZEN 3500 (NanoMalvern Panalytical, UK) using a transparent cuvette. NPs morphology and shape were analyzed by transmission electron microscopy (TEM) using a LIBRA 120 PLUS TEM (Carl Zeiss NTS GmbH, Oberkochen, Germany) and analyzed with Xei data acquisition software.

#### 2.2.5. Stability Study of **JE22-NPs** (**5**)

For the stability study, 10 µL of NPs were incubated for 24 h in ultrapure water (Milli-Q grade, H_2_O mq), DMEM, NaCl 10 mM, NaCl 154 mM, and PBS at pH = 7 at 4 °C and 37 °C. Then, NPs were centrifuged and prepared in biological grade water, and subsequently, the particle mean size and size distribution were determined by DLS and zeta potential analysis. 

#### 2.2.6. Determination of Conjugation Efficiency of **JE22-NPs** (**5**) 

Calculation of **JE22** conjugation efficiency (CE; %) and loading capacity (LC) was carried out by measurement of the concentration of free **JE22** in the supernatant obtained after the centrifugation of NPs by UV spectroscopy at 254 nm. Previously, an absorbance study of **JE22** at different concentrations and a calibration curve with lineal ratio between **JE22** concentration and the optical density of the compound was performed (Appendix A). Subsequently, **JE22** LC and CE were calculated based on formulas as follows:(1)$$LC=\frac{\left[JE22\;conjugated\;on\;nanoparticle\;surface\right]}{Number\;of\;NPs}\times N_{A}$$
where *N_A_* is Avogadro’s number.
(2)$$CE\;\left(\%\right)=\frac{\left[JE22\;conjugated\;on\;nanoparticle\;surface\right]}{\begin{array}{c} Loading\;of\;free\;amine\;groups\\ on\;nanoparticle\;surface \end{array}}\times 100$$

#### 2.2.7. Evaluation of Drug Release Profile of **JE22-NPs** (**5**) 

To determine the efficiency of the hydrolysis of the hydrazone bond of the **JE22-NPs** (**5**), samples at acidic and neutral pH were prepared. First, 200 µL (8.81 × 10^8^ NPs/µL) of NPs were incubated in a PBS solution at pH = 5 and pH = 7 for 120 h in an incubator at 37 °C. Then, the supernatants were collected by centrifuging each sample at *t* = 1.5, 3, 6, 24, 48, 72, 96, and 120 h, and they were analyzed through high-performance liquid chromatography (HPLC) (Agilent 1200 series HPLC system) with a C18 column from Waters CORTECS™ (2.1 mm × 100 mm, 1.6 μm) [21]. The detection of PDA eλ for **JE22** was established at 252 nm. The mobile phase of water (0.1% formic acid): acetonitrile was supplied at a flow rate of 0.4 mL/min: 0% B, T8: 95% B, T8.1: 0% B, analysis time 10 min. Using standard samples, a calibration curve of **JE22** was prepared (Appendix A). The maximum identification was confirmed by the retention time (RT) of **JE22** at 1.55 min. Cumulative release of **JE22** was performed using the following equation:(3)$$Cumulative\;JE22\;Release\;\left(\%\right)=\frac{D_{t}}{D_{T}}\times 100$$
where *D_T_* is the total concentration of **JE22**-loaded onto the **JE22-NPs** (**5**), and *D_t_* is the concentration of **JE22** released from **JE22-NPs** (**5**) at a given time *t* [21].

### 2.3. Biology

#### 2.3.1. General

A NuAire NU-4750E US AutoFlow incubator was used for cell culture. Cell-based experiments were carried out in a TELSTAR BIO II Class II A laminar flow cabinet. Flow cytometry assays were performed on a FACSCanto II system (Becton Dickinson & Co., Franklin Lakes, NJ, USA) using the Flowjo^®^ 10 software (Becton Dickinson & Co., Franklin Lakes, NJ, USA) for analysis. Transmission electron microscopy was performed on a LIBRA 120 PLUS Carl Zeiss SMT microscope. Cell viability was carried out using a GloMax-Multi Detection System to measure fluorescence. Wound healing images were acquired using an Olympus CKX53 microscope, and wound areas were measured using ImageJ^®^ software (version 1.49b, Rasband, W.S., U. S. National Institutes of Health, Bethesda, MD, USA). Confocal microscopy images were obtained using a Zeiss LSM 710 confocal laser scanning microscope and ZEN 2012 program Blue Edition (Carl Zeiss NTS GmbH, Oberkochen, Germany) for image acquisition. 

#### 2.3.2. Cell Culture

Human breast carcinoma MDA-MB-231 and MCF-7 cells and human embryonic kidney-derived non-cancerous cells HEK-293 (provided by the Cell Bank the Center of Scientific Instrumentation of the University of Granada) (obtained from American Type Culture Collection ATCC, Manassas, VA, USA) were cultured in DMEM with serum (10% FBS), L-glutamine (2 mM), and 1% penicillin/streptomycin and incubated in a tissue culture incubator at 37 °C, 5% CO_2_ and 95% relative humidity. Cells were frequently tested negative for mycoplasma infection.

#### 2.3.3. Cell Viability Assays

**JE22** was dissolved in DMSO and stored at −20 °C. For each experiment, the stock solution (100 mM) was further diluted in culture media to obtain the desired concentrations. MDA-MB-231 cells were seeded in a 96-well plate format (1000 cells/well) and incubated for 24 h before treatment. Each well was then replaced with fresh media, containing **JE22** (0.01–100 µM) and incubated for 5 days. Untreated cells (DMSO, 0.1% *v*/*v*) were used as control to detect any undesirable effects of culture conditions on cell viability. Each condition was performed in triplicates. PrestoBlue^TM^ cell viability reagent (10% *v*/*v*) was added to each well and the plate incubated for 120 min. Fluorescence emission was detected using a GloMax-Multi Detection System (excitation filter at 540 nm and emission filter at 590 nm). All conditions were normalized to the untreated cells (100%) and the curve fitted using GraphPad Prism using a sigmoidal variable slope curve. The EC_50_ (half-maximal effective concentration) value is expressed as the mean ± SD of three independent experiments.

For viability assays of **JE22-NPs** (**5**), MDA-MB-231 and HEK-293 cells were plated at 1000 cells/well (doubling times are 26 h and 24 h, respectively) and MCF-7 cells were plated at 2000 cells/well (doubling time is 34 h). After 24h, cells were nanofected with different ratios of **JE22-NPs** (**5**) (40,000, 20,000, 10,000, 5000, 2500, 1250, and 625 NPs/cell). Untreated cells, cells incubated with **Naked-NPs** (**1**) (40,000 NPs/cell), and NPs in culture medium in the absence of cells were used as controls. Each condition was performed in triplicate. Cell viability was tested at day 5 using PrestoBlue^TM^ reagent and curve fitted as previously described. For viability assays of **JE22-NPs** (**5**) at acidic conditions, cells were treated with 40,000 NPs/cell in DMEM media at pH = 5. DMEM media at pH = 5 were prepared by replacing sodium bicarbonate with PIPES buffer (10 mM) and adjusting the pH with NaOH. Untreated cells, cells incubated with **Naked-NPs** (**1**) (40,000 NPs/cell), and NPs in culture medium in the absence of cells were used as control. After 1.5 h of incubation, media were replaced with pH = 7.4 DMEM media and cell viability was tested at day 5 as described above. Each condition was performed in triplicates.

#### 2.3.4. Confocal Microscopy Analysis

Glass coverslips were coated with poly-L-lysine (10 × 10^4^ cells/well), and then MDA-MB-231 cells were seeded onto them in 24-well plate format. Following incubation time (24 h), cells were stained using an anti-CD44-FITC antibody diluted in MACS^®^ BSA Stock Solution (1 µL/400 µL, Miltenyi Biomedicine GmbH, Bergisch Gladbach, Germany). Plates were incubated for 10 min on ice in the dark. Then, cells were washed with DMEM media and treated with a new solution of culture media containing **JE22-NPs** (**5**) fluorescently labeled with Cy5 (1000 NPs/cell). After 30 min of incubation at 37 °C, the medium was aspirated, and the cells were washed twice with 1 × PBS and fixed in 4% paraformaldehyde for 10 min at room temperature. After washing with 1 × PBS, fixed cells were mounted with DAPI-containing mounting medium (ProLong Gold, Life technologies, Renfrew, UK). A ZEISS LSM 710 confocal laser microscope (Carl Zeiss NTS GmbH, Oberkochen, Germany) was used to collect the images using a DIC Plan-Apochromat 63× oil immersion objective with 1.40 numerical apertures and the ZEN 2010 software (Carl Zeiss NTS GmbH, Oberkochen, Germany). Images were subsequently analyzed with both the Zen 2012 Blue Edition Image and ImageJ softwares (version 1.49b, Rasband, W.S., U. S. National Institutes of Health, Bethesda, MD, USA).

#### 2.3.5. HA-FITC Binding Assay

Adherent MDA-MB-231 cells were trypsinized, counted, and diluted in DMEM in order to have 5 × 10^4^ cells/eppendorf tube. Cells were centrifuged for 5 min, and pellets were resuspended in DMEM media containing **JE22-NPs** (**5**) (40,000 NPs/cell) or **JE22** in solution (120 µg/mL). Anti-CD44 antibody was used as control (120 µg/mL). Samples were incubated at 4 °C for 30 min. Then, cells were centrifuged for 5 min, and pellets were resuspended in DMEM media containing HA-FITC (20 µg/mL) and incubated at 4 °C for 15 min. Cells incubated with unlabeled HA were used as the negative control, whereas cells incubated with HA–FITC served as the positive control. After incubation, cells were centrifuged and resuspended in PBS, and samples were analyzed by flow cytometry (FACSCanto II, Becton Dickinson & Co., Franklin Lakes, NJ, USA). Flowjo^®^ 10 software (Becton Dickinson & Co., Franklin Lakes, NJ, USA) was used for data analysis. Results are expressed as the fluorescence intensity mean ± SD of three independent experiments.

#### 2.3.6. Wound Healing Assay

MDA-MB-231 cells were seeded in a 12-well plate format at 25 × 10^4^ cells/well and incubated until 90% confluence. Then, cells were gently scratched using a pipette tip, washed with PBS to remove cell debris, and treated with **JE22-NPs** (**5**) at 20,000 NPs/cell (36 nM) and **JE22** (EC_50_ = 8 µM). Untreated cells (DMSO, 0.1% *v*/*v*) and cells treated with **Naked-NPs** (**1**) at 20,000 NPs were used as controls. Images were acquired at time zero and after 24 h of incubation using an Olympus CKX53 microscope (4× objective magnification). Wound areas were measured using ImageJ^®^ software (Rasband, W.S., U. S. National Institutes of Health, Bethesda, MD, USA).

#### 2.3.7. Apoptosis Assay 

MDA-MB-231 and MCF-7 cells were seeded at 5 × 10^4^ cells/well in a 24-well plate. After 24 h, cells were treated with **JE22-NPs** (**5**) at 20,000 NPs/cell (36 nM), **JE22** (EC_50_ = 8 µM and 4 × EC_50_ = 32 µM), and **Naked-NPs** (**1**) (20,000 NPs/cell) for 24 h. Cells incubated in the absence of the apoptosis inducing agent were used as the negative control, whereas cells incubated with H_2_O_2_ (2 mM) for 4 h at 37 °C served as the positive control. The experiments were performed using the Annexin V-FITC detection kit (Tali Apoptosis Kit -Annexin V Alexa Fluor 488 and propidium iodide (A10788, Invitrogen Europe Limited, Renfrew, UK)) according to the manufacturer’s instructions, and the samples were analyzed by flow cytometry with a FACSCanto II flow cytometer. Flowjo^®^ 10 software was used for data treatment. The analysis was performed in three independent assays.

#### 2.3.8. Statistical Analysis

A one-way analysis of variance (ANOVA) was performed using Sigmastat 3.5 statistical analysis software. 

## 3. Results and Discussion

### 3.1. Study of the Effect of Structural Modification of THIQ on CD44 Interaction by Computational Studies

To assess whether the functionalization with ketone does not affect the interaction with the CD44, a computational study was performed. For this purpose, the protein–ligand interactions between the murine CD44 HABD and **JE22** at an atomist level were analyzed. MD simulations using the crystallized CD44 HABD with **THIQ-ester** derivative were performed using the same protocols after substituting one residue by the other (Figure 2A,B). A typical setup of the MD simulations with an explicit solvent is shown in Figure 2A for the protein with one of the ligands. 

In order to further characterize the multiple poses that we observed in the MD trajectories of either ligand, we next performed a cluster analysis using principal component analysis (PCA) and the distances between (i) the center of mass of the N and O atoms of **JE22** or **THIQ-ester** and (ii) the center of mass of N and O atoms of residues Asn29, Thr31, Thr80, Cys81, Arg82, and Arg155. These residues are those that have at least one atom within 3 Å from the ligand at the beginning of the simulation. The data were filtered, and only when the minimum distance between the protein residues and either **JE22** or **THIQ-ester** was less than 3 Å, they were kept. Subsequently, a PCA was performed over the resulting multidimensional matrix of distances. 95% of the variance of the data was explained with the sum of the eight first principal components (PCs) (Figure 2C,D). Clustering was just performed with only the first two PCs that amounted to ~70% of the variance. This analysis was performed with a Kmeans algorithm using four clusters as initial guess; two of the clusters were rather localized and the others two were disperse (Figure 2E,F). 

These computational studies have demonstrated that **JE22** binds to murine CD44 HABD in an almost identical fashion than **THIQ-ester** derivative, which is found in the reported crystallographic structure [16]. Based on these results, we can confirm that the designed THIQ ketone derivative **JE22** is a good candidate to interact with CD44 receptor.

### 3.2. Synthesis and Physicochemical Characterization of the Nanodevice to Target CD44

#### 3.2.1. Preparation of **JE22-NPs** (**5**)

Following a previously described protocol, a monodisperse population of amino-functionalized polystyrene nanoparticles cross-linked with divinylbenzene were synthesized by dispersion polymerization, using vinyl benzyl amine hydrochloride—VBAH—as the monomer to functionalize the nanoparticle with the amino groups [17]. Following an Fmoc solid-phase protocol, **Naked-NPs** (**1**) (0.064 mmol g^−1^ of amino groups) were PEGylated to obtained **PEGylated-NPs** (**2**) (100% yield). The PEGylation increases the biocompatibility of the NPs and reduces unfavorable interactions between NPs and the bioactive cargoes. The modified THIQ derivative of CD44 inhibitor (**JE22**) was synthesized as described in Figure 1. Ketone moiety to allow for conjugation to the nanoparticle by hydrazine formation was achieved by the Michael addition of THIQ to methyl vinyl ketone, employing cupper bromide (I) as a catalyst (Figure 1). The structure of the obtained compound **JE22** was confirmed by NMR and mass spectra (Appendix A). Then, drug loading was carried out by conjugation of the CD44 inhibitor **JE22** via hydrazone bond [20]. For this purpose, carboxylated nanoparticles **COOH-NPs** (**3**) were prepared using succinic anhydride; then, **hydrazine-NPs** (**4**) were prepared by treatment with hydrazine, and the selective conjugation to the ketone group of THIQ derivative **JE22** was carried out to yield **JE22–NPs** (**5**) (Figure 1).

#### 3.2.2. Physicochemical Characterization of **JE22-NPs** (**5**)

The size distribution of the nanoparticles loaded with THIQ ketone derivative **JE22-NPs** (**5**) and **Naked-NPs** (**1**) were measured by DLS (Figure 3A). A monodisperse population was observed with a hydrodynamic diameter of 382.5 ±0.9 nm (PDI = 0.13) (Figure 3A). TEM analysis revealed the spherical shape of these nanoparticles and corroborated their size (Figure 3D). The zeta potential of **JE22-NPs** (**5**) and **Naked-NPs** (**1**) was also determined. The value for the new nanoformulation was slightly negative (−24.1 mV ± 0.7) in water (Figure 3B). 

Next, the stability of **JE22-NPs** (**5**) was evaluated in different conditions at 4 °C and 37 °C following guidelines provided by the European Nanomedicine Characterization lab. The size of these nanodevices was measured by DLS after 24 h in several sterile media: ultrapure water (Milli-Q grade, H_2_O mq), DMEM, NaCl 10 mM, NaCl 154 mM, and PBS pH = 7, showing a constant size distribution (Figure 3C). It was observed that neither the temperature nor the composition of the vehicle affected the stability of these nanoparticles. These results were corroborated by the zeta potential analysis (Appendix A). Overall, the stability of these particles was confirmed, which is a key point for future translation of this nanodevice. 

#### 3.2.3. Efficiency of Conjugation and Drug Release of **JE22-NPs** (**5**)

The quantification of the remaining amount of drug in the supernatant of the reaction can give information about the efficiency of the conjugation of anti-CD44 derivative **JE22** to the nanoparticles. For this purpose, a calibration curve of **JE22** was generated measuring a set of standard samples by UV spectroscopy (A_254_ nm) (Appendix A). Then, the LC value was determined by considering the amount of conjugated **JE22** with respect to the number of nanoparticles, and this approach is more accurate than nanoparticle weight [35]. To this aim, the number of nanoparticles per volume was determined using an accurate spectrophotometric method that was previously developed [36]. The concentration of nanoparticles **JE22-NPs** (**5**) was estimated as 4.8 × 10^6^ NPs/mL (Appendix A). The drug LC is related to the number of nanoparticles; thereafter, the LC per nanoparticle can be calculated. A LC of 1.14 × 10^7^ molecules of **JE22** per nanoparticle was estimated. This value corresponds to 1.89 × 10^−8^ nmol of CD44 inhibitor **JE22** per NP (Figure 4A). 

To determine the value of drug dose with accuracy and precision is of extreme relevance for the clinical translation of nanomedicine. The CE was determined by considering the drug conjugated with respect to the total amount of free amine groups on the nanoparticle surface, which was 100% for **JE22-NPs** (**5**) (Figure 4A). This high efficiency is remarkable compared to drug-loading strategy based on encapsulation [37].

To achieve selective release at the target site, a pH-sensitive stimuli discharge strategy was implemented. Based on the fact that there is slightly acidic pH at the tumor microenvironment, a cleavable bond that responds to acidic pH was implemented [38]. To release the drug in acidic conditions, **JE22** was covalently conjugated to nanoparticles by a hydrazone bond sensitive to pH = 5–6 as previously reported [35]. The pH-responsive release of THIQ derivative **JE22** in vitro was determined by HPLC. A comparison of the percentage of the released drug with respect to the amount of CD44 inhibitor conjugated to the **JE22-NPs** (**5**) was done. Release profiles were obtained for five days at pH = 5 and pH = 7.4 by HPLC analysis (Figure 4B). As expected, the pH-sensitive cleavage of the hydrazone linker resulted in the exponential sustained release of **JE22** in an acidic environment (pH = 5 PBS) (Figure 4B, blue line). An accumulative release was achieved for up to 5 days (120 h). A significant release was observed within 6 h of incubation at pH = 5 with a release rate of 78% ± 0.6. Then, the maximum release value of 100% was achieved by a sustained release for up to 120 h. Non-significant size change was observed following incubation at pH = 5 for this period (see Appendix A). On the other hand, the amount of drug released from the nanodevice was minimal at physiological environment (pH = 7.4 PBS), (~20% within 6 h of incubation). This result demonstrates that a significant amount of the drug remained attached to the nanoparticles (Figure 4B, orange line). This result pointed out the realistic stability and selectivity of the **JE22-NPs** (**5**). It is important to remark that the pH value of the medium has a clear effect on the release efficiency of the drug, which validates the drug release strategy designed in this approach. Then, the efficient release of the drug in a sustained manner in acidic conditions could be a key feature to improve the therapeutic efficacy of **JE22-NPs** (**5**) in the tumor site. 

### 3.3. Evaluation of Efficiency of the Designed Nanodevice **JE22-NP** (**5**) for Antitumor Activity 

#### Assessment of Biological Activity of **JE22-NPs** (**5**)

In order to assess the biological activity of this nanodevice to target CD44, in agreement with previous studies [39,40], two breast cancer cell lines expressing different levels of CD44 were selected: MDA-MB-231 with a high level and MCF-7 with a low level of CD44 expression, respectively. Analysis of CD44 expression by flow cytometry using an anti-CD44 antibody labeled with fluorescein (anti-CD44-FITC) confirms the suitability of these cell lines for testing CD44 inhibition (Appendix A).

The half maximal efficacy concentration (EC_50_) of this therapeutic nanodevice **JE22-NPs** (**5**) in MDA-MB-231 cells was determined. For this purpose, cell viability was monitored using fluorescent resazurin assay. EC_50_ values were calculated from the generated 10-point semilog dose–response curves (Figure 5A–C). Initially, MDA-MB-231 cells were treated for 120 h with increasing concentrations of **JE22** in solution (0.001 to 100 µM) to determine the range of doses of inhibitor required to achieve the antiproliferative activity. Free **JE22** has an EC_50_ value of 8 µM in MDA-MB-231 cells (Figure 5A). Then, a range of different concentrations of **JE22-NPs** (**5**) (312–40,000 NPs/cells, that corresponds to 0.6–72 nM) were incubated for 120 h with MDA-MB-231 cells. The EC_50_ value for therapeutic NPs (**JE22-NPs, 5**) was calculated to be 49 nM (Figure 5C), which corresponds to 27,367 NPs/cell (Appendix A). This value indicated that the nanosystem offers a 150-fold reduction of the amount of **JE22** required to have the same effect than the free form has in tumor cells overexpressing CD44. In addition, treatment of CD44 low-expression MCF-7 cells with **JE22-NPs** (**5**) show no significant reduction of cell viability (Appendix A), reinforcing the selective effect of the nanodevice against CD44. 

To further verify the selectivity of the nanodevice targeting the CD44 receptor in the MDA-MB-231 cancer cell line, we performed a competitive binding experiment. Cells were preincubated with the anti-CD44 antibody before treatment with **JE22-NPs** (**5**), showing a significant decrease of the antiproliferative effect respect to cells without pretreatment (Appendix A). These results showed that pretreating cells with the antibody effectively blocked CD44 cell-binding sites, preventing the recognition of epitopes from the nanodevices, showing that therapeutic effect of **JE22-NPs** (**5**) is linked to CD44 recognition.

Based on the fact that tumor tissues are characterized by an acidic extracellular pH as a result of the altered cancer cell metabolism compared to normal tissues, we applied a chemical strategy to achieve the release of the drug from the nanodevice in acidic conditions. To mimic the acidic tumor microenvironment, we used bicarbonate-free DMEM buffered with 10 mM of PIPES to fix a slightly more acidic external medium. We first tested whether cell viability could be affected after the incubation of MDA-MB-231 cells with pH = 5 DMEM media. After 1.5 h of incubation, no signs of cell death were observed, although a significant reduction of cell viability was obtained after 3 and 6 h of incubation (Appendix A). In order to test the effect of this selective release in the antiproliferative effect, we performed a comparative experiment incubating the cells in standard conditions (DMEM medium, pH = 7.5) and in the presence of DMEM buffered with PIPES to guarantee a slightly more acidic external medium (pH = 5). The results indicate that cytotoxic activities have pH dependence. Remarkably, **JE22-NPs** (**5**) showed to be more cytotoxic at acidic extracellular pH = 5 following only 1.5 h of incubation compared to under conventional conditions (pH = 7.4) for 5 days (see Figure 5D). This result is in agreement with the maximum peak of release of compound by HPLC analysis. A sustained drug release under physiological conditions can occur due to the acidification of culture media overtime [41]. As expected, the pH does not have any significant effect in cell viability when cells are treated with **Naked-NPs** (**1**)**.** These results suggest that specific release in acidic conditions is crucial for the therapeutic activity of this compound.

Finally, the cytotoxic effect of this nanodevice for non-cancerous cells was evaluated. We have used the human embryonic kidney-derived non-cancerous cells (HEK-293) to analyze the cell viability of **JE22-NPs** (**5**). No sign of death was observed after treatment of the normal cells with the nanodevice (Appendix A). 

Overall, the nanoparticles were toxic to CD44+ cells and non-toxic to CD44- and non-cancerous cells.

### 3.4. Evaluation of Efficiency of the Designed Nanodevice for the Inhibition of CD44 Receptor Binding 

#### 3.4.1. Analysis of the Interaction of the Designed Nanodevice **JE22-NPs (5)** with CD44+ Cells by Confocal Microscopy 

A confocal microscopy approach was carried out to study the location of the nanoparticles loaded with **JE22** on the surface of the CD44+ cells. MDA-MB-231 cells were immunolabeled with a fluorescently tagged primary anti-CD44 antibody (anti-CD44-FITC) [42]. Then, cells were treated with **JE22-NPs** (**5**) labelled with a cyanine derivative (Cy5, excitation 651 nm and emission 670 nm) to track them by fluorescence microscopy (see protocol for fluorescent labelling in Appendix A). It can be observed that the extracellular location of the **JE22-NPs** (**5**) is on the cellular surface where CD44 receptor is expressed (Figure 6 and Appendix A). 

#### 3.4.2. Assessment of CD44-Binding Capacity

To assess the CD44-binding capacity of the THIQ derivative **JE22** in solution and conjugated to the nanodevice, we performed a competitive binding assay using a fluorescent-labeled derivative of HA as natural ligand of CD44 (HA-FITC), which has high capacity of binding the CD44 receptor. For this purpose, we followed a previously reported method with slight modifications [43]. Briefly, cells overexpressing CD44, MDA-MB-231 cells, were pre-incubated with **JE22** and **JE22-NPs** (**5**) at 4 °C, and physiological pH for 30 min to allow for their binding to CD44 receptor on the cell surface. Then, cells were incubated with HA-FITC at 4 °C for 15 min, and fluorescence analyzed by flow cytometry. Anti-CD44 antibody was used as positive control. Cells incubated with unlabeled HA were used as negative control. As observed in Figure 7, pre-incubation with **JE22** displaced HA-FITC binding, yielding a statistically significant reduction of the fluorescence intensity (1.3-fold reduction) compared to the cells incubated with HA-FITC and cells pretreated with anti-CD44 monoclonal antibody (2.20-fold reduction). It is remarkably the fact that, as we expected, when cells were pretreated with the designed nanodevice **JE22-NPs** (**5**), no significant displacement was observed. This result reinforces our initial hypothesis that brought us to design a pH-sensitive strategy to release the CD44 inhibitor from the nanodevice at the acidic tumor microenvironment to enhance the efficient interaction with the CD44 receptor and to significantly reduce HA binding.

#### 3.4.3. Influence of **JE22-NPs** (**5**) in Migration of CD44+ Cells

Cell migration participates in numerous physiological and pathological processes. Previous studies have shown that CD44 proteins can stimulate tumor cell proliferation, motility, and invasion [44]. 

To detect whether this nanodevice can decrease the migration of CD44+ cells and, consequently, modify any mesenchymal behavior, a scratch-wound healing migration assay was performed to determine whether **JE22-NPs** (**5**) compared to free **JE22** could halt migration of MDA-MB-231 cells, as would be expected for a CD44 inhibitor [45]. After making the wound, MDA-MB-231 cells were treated with CD44 inhibitor **JE22** free and conjugated to nanodevice **JE22-NPs** (**5**) for 24 h and compared with untreated cells (0.1% *v*/*v* DMSO) and cells treated with the nanodevice without drugs (**Naked-NPs, 1**), respectively. Cells treated with THIQ derivative **JE22** in solution significantly reduced cell motility; wound closure was reduced to 15.09 ± 4.90% compared with cells treated with DMSO (30.21 ± 6.06%). Remarkably, when MDA-MB-231 cells were treated with **JE22** conjugated to the nanodevice **JE22-NPs** (**5**), cell migration reduction was significantly higher than with **JE22** in solution (4.89 ± 2.77%) (Figure 8). The fact that the concentration used in this assay is lower than the EC_50_ (20,000 NPs) (Figure 5B) suggests that the migration effect could be independent of the cytotoxic effect.

#### 3.4.4. Apoptotic Activity of **JE22-NPs** (**5**)

To rule out that the observed delay in gap closure was caused by **JE22**-induced cell death, cell viability was examined by Annexin V/PI staining after treatment. Annexin V/PI staining is a widely used method to study apoptotic cells. Annexin V/PI significantly regulates the viable, necrotic, and apoptotic cells through differences in plasma membrane permeability and integrity [46]. We first study the apoptotic effect of **JE22** against CD44+ MDA-MB-231. Even at four times the EC_50_ concentration of the drug, no crucial apoptotic effect was observed (Figure 9). These results suggest that apoptosis is not the mechanism by which **JE22** induces cell death. Then, the apoptotic effect of the **Naked-NPs** (**1**) and **JE22-NPs** (**5**) was analyzed to check that no apoptotic effect is caused by the nanocarrier itself. As expected the concentration of nanoparticles **JE22-NPs** (**5**) that we used in this assay (20,000 NPs) did not induce significant apoptosis or necrosis in either MDA-MB-321 cells (Figure 9) or MCF-7 (Appendix A). However, gap closure was inhibited at the same concentration (Figure 8). These results reinforce the fact that the migration effect could be independent of the cytotoxic effect of **JE22-NPs** (**5**) [47].

## 4. Conclusions

In this work, we successfully designed and evaluated an innovative nanodevice for selective anticancer therapy targeting CD44 receptors. In particular, a ketone derivative of THIQ (**JE22**) to target the CD44 HABD was synthetized and conjugated to polymeric nanoparticles via hydrazone bond to achieve a nanodevice for selective release in tumor microenvironments. Computational analysis confirmed that the designed THIQ ketone derivative (**JE22**) is a good candidate to interact with the CD44 receptor. Remarkably, the conjugation of this CD44 inhibitor to the nanodevice **JE22-NPs** (**5**) achieved more than 150-fold reduction of the dosis required to render a significant therapeutic effect. It was determined that the pH-sensitive strategy to release the CD44 inhibitor from the nanodevice at the acidic tumor microenviroment enhanced the efficient interaction with the CD44 receptor and significantly reduced HA binding. The migration of CD44+ cells decreased, and a non-significant apoptotic effect was observed following treatment with **JE22-NPs** (**5**). Additionally, the nanodevide was selective to cancerous CD44+ cells and safe to non-cancerous cells.

Based on this preliminary investigation, further studies will be undertaken to characterise the interaction between this THIQ derivative and the CD44 HA binding domain to design a next generation of CD44 targeted nanotherapies.

## Data Availability

The data that support the findings of this study are freely available on request from the corresponding author.

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
