# Peer review of "Selective Anticancer Therapy Based on a HA-CD44 Interaction Inhibitor Loaded on Polymeric Nanoparticles"

_pharmaceutics, 2022, doi:10.3390/pharmaceutics14040788_

Round 1

Reviewer 1 Report

Comments

  1. Line 413-414: As the value of zeta potential of these nanoparticles is above −20 mV, it can be predicted that their stability won´t be compromised” any reference? In general, it is considered that the absolute value of ZP should be around 30 mV for any stable colloidal dispersion.
  2. X-axis of Figure 5A, is it only concentration or Log Concentration? X-axis of Figure 5C, is it Log Concentration of the therapeutic agent/drug or the nano-carrier? X-axis of Figure 5B. what does it NPs/cell mean? How the NPs/cell was calculated?
  3. Line 538: “A range of nanoparticles from 312 to 40,000 was tested”, how this range was determined? Figure 5D: “Therapeutic effect representation by bar graph of JE22-NPs (5) at pH= 7.4 versus cell viability at pH= 5”, what was the duration treatment?
  4. Mention the doubling time of MDA-MB-231 cells? Cell viability was determined after 5 days’ treatment, right. Why not the viability of cells were determined at 24 h, 48 h, 72 h and at 5th day?
  5. Line 587-588: “Mean fluorescence of MDA-MB-231 cells analysed by flow cytometry after treatment with JE22-NPs (5)” what was the duration treatment?
  6. Line 632: how the 20000 NPs/cell was estimated/ determined/ calculated?
  7. The stability of JE22‐NPs (5) was evaluated in different standard storage conditions and the size of these nano-devices were measured by DLS after 24 h at 4°C and 37°C only, why the stability was done for 24 h only? By which guideline? Along with size, zeta-potential should also be determined.
  8. No results were discussed about TEM analysis as shown in Figure 3D?
  9. The unit of time should be mentioned as “h” rather than “hours”.
  10. Line 456: “The pH‐responsive release of THIQ derivative JE22 in vitro was determined by 456 HPLC”, the chromatographic condition should be mentioned here (not in the Supplementary) with proper reference(s).
  11. Either in abstract or introduction, at least once the THIQ should be described before putting its abbreviation at all places.
  12. What were the doubling time of MDA-MB-231 and MCF-7 cells? Why not the cell viability was checked at 24h, 48h, 72 h and at day 5? Why it was tested only at 5th day of incubation?

Reviewer 2 Report

I revised the manuscript Pharmaceutics-1647824 entitled “Selective anticancer therapy based on a CD44-HA interaction inhibitor loaded on polymeric nanoparticles” by Román et al.

I read this manuscript very carefully and I think the authors presents a very well organized manuscript with many assays to prove the noteworthiness of the device JE22-NPs in the treatment of cancer. However, the organic compound they bind to nanoparticles is already described in the literature so the work loses in terms of its innovation.

I leave below a set of notes that the authors should take into account in order to improve this paper.

Given the above, I am of opinion that this paper should be accepted after the suggested corrections.

Line

Comments

21

The acronym HA-CD44 does not agree with the acronym in the title of the article (CD44-HA), so the authors must replace it to be the same throughout the manuscript.

22

Like the others acronyms in abstract, the acronym THIQ must have its meaning in parentheses.

93

The authors placed the number (5) in JE22-NPs (5) in the figure caption, but it should also appear in the figure.

151

This compound has already been synthesized and fully characterized by Jones et al. [33], so it is not a new compound. As such, the authors should take the synthesis and characterization of JE22 from here and place it in the supplementary material. The authors should mention at this point that the compound was synthesized according to the method mentioned by Jones et al. [33] and that the spectroscopic data are in agreement with those reported in the literature [33].

154

Change "ml" to "mL" and “l” to “L “throughout the manuscript.

168

Put the full point after the reference  ….tails). [17]

169

The multiplicative symbol in (1 mL x 3 times) is uncorrect. So the authors have to change the symbol “x” to symbol “×” used in (3 × 20 min) on line 175, throughout the manuscript.

178

Change this sentence “…this solution was added to NPs and mixed..” to…”and this last solution mixed to NPs…”

184

“stirring for 25”… change to …”stirring at 25”

189

The scheme 1 should be mentioned at the beginning of this paragraph for a better understanding of the reader.

223

Delete the comma in “S9C,)”

240

The authors must refer the percentage of antibiotic that was used for cell culture.

251

The authors refer that cells were incubated with JE22 compound for 5 days. In my experience it is too much time without replacing media, since cells comes into senescence, compromising the viability cell results. The authors must explain if a control was made in order to know if 5 days without replacing media does not had any effect on the cell culture.

276

Please insert the term and in the sentence …”cells/well, and then”

293

At the end of this line replace the word “with” by “in”

303

Change the term mean in “mean fluorescence intensity ± SD” to “fluorescence intensity mean ± SD”

379

Remove the parenthesis in “(using vinyl benzyl a…” and add a comma after the word polymerization

381

Remove the parenthesis in “groups) [17].

389

Change RMN by NMR

390

Put the full stop after the parentheses in… “bond. [20]”

399

In section 2.2.3 the authors indicated equivalents without full stop (eq) but sometimes they use “eq.”. The last one is correct so, please change it throughout the manuscript.

417

Change the word …“solutions:” to “conditions:”

504

 Delete a full stop at the end of sentence.

558

The scales in Figure 6 are not visible, so the authors should improve the images

562

Please change  “anti-humanCD44” to “anti-human CD44”

577

The authors report that there is a significant reduction in fluorescence intensity. In my view the reduction of 1.3 compared to 2.2 is not very significant, so the authors should rephrase the sentence.

613

For the caption to be consistent with the images, it is necessary to place the numbers of the compounds in the images as they are in the caption.

619

In this paragraph the authors are not consistent in the designations they present, using the terms AnxV/PI, Annexin/PI, and Annexin V in conjunction with propidium iodide. They have to standardize.

623

Authors must indicate a reference to justify the …”permeability and integrity.”

630

The authors should make a more exhaustive interpretation of the results shown in Figure 9 and should show the pH at which the tests were carried out because if they were carried out at pH 7, it was already expected that significant cell death would not occur.

Author Response

Please the attachment

Reviewer 3 Report

The work on new nanocarriers of therapeutics for anti-cancer therapies is an interesting direction of research. The work is consistent, the authors have found a balance in interdisciplinarity. The results are interesting, but NPs cytotoxic analysis for normal cells is lacking. The toxicity of this component for non-cancerous cells could make the proposed biomedical implications only a theory.

Round 2

Reviewer 1 Report

Authors have responded the comments properly.

This manuscript is a resubmission of an earlier submission. The following is a list of the peer review reports and author responses from that submission.

Round 1

Reviewer 1 Report

I think that this paper is very well written and results well explained. I have only some suggestions and critics regarding cell biology experiments: - confocal microscopy: I think it could be interesting make some images of cells treated at 37°C given that experiments of cell viability and apoptosis were done at 37°C. Moreover, the real condition in human body is under 37°C. Furthermore, it could be useful to see other images to be able to affirm the exact location of nanoparticles. - the conclusion of 3.3.3. paragraph is weak. tha author suggested that their results reinforce their hypothesis regarding pH sensibility. Howevere I think that if they want to confirm this hypothesis they have tot do the same experiments at pH5 as well as for cell viability assay. How they expleained the only half-reduction of HA-FITC binding to the cells after treating cells with antibody for CD44? - I don't undesterdand why authors decided to use 20000 MPs for migration (maybe because at 40000 they are too toxic?) and apoptosis. Mostly for apoptosis assay I think it could be useful to compare the results obtained for cell viability, so the experiment have to be done with the same NPs concentration. - conclusion: to affimr that they have designed a selective anticancer therapy they have to perform experiments also with CD44 negative cells

Reviewer 2 Report

Comments to the Authors

The manuscript 1512392 titled “Selective anticancer therapy based on CD44 inhibitors loaded nanoparticles”, focus on an interesting field, describing an innovative antitumor treatment strategy based on the development of a nanodevice (JE22- Aminomethyl NPs) for selective release of THIQ-ketone derivative (JE22) an inhibitor of the HA-CD44 interactions.

The introduction provides sufficient background and includes all relevant and also recent references; the experimental design is clear and well written, and the results are well presented in the text.

In any way, minor revisions are recommended.

Some suggestions to improve a biological aspect of this study, about the effects of the synthesized nanodevice on tumor cell proliferation. In fact, as you reported in line 557, the CD44 membrane receptor can stimulate cell proliferation, motility, and invasion. In your contribution, you show the influence of JE22-NPs treatment on the migration and viability of human breast cancer CD44+ cells. Although, another main and peculiar aspect of tumor malignancy is the proliferation activity of cells, which could be changed after anticancer treatment. Thus, it will be necessary to analyze also the behavior of MDA-MB-231 cells in terms of proliferation activity after treatment with JE22-NPs at different times, to have a complete and exhaustive idea of your proposed selective anticancer therapy.

Minor concerns:

-Better specify the title, which is general and somewhat pretentious. I’d talk about “Selective anticancer therapy based on a CD44-HA bind inhibitor loaded on polymeric nanoparticles

-Line 151…: should be reported in a table of paragraph 2.2.3 the meaning of bold numbers between brackets

-Line 251: why it establish 1,5h as the time of incubation? It should be interesting also valuate the time kinetics of cell viability after JE22-NPs treatment.

Line 509: there ia a repeat. I suggest to change “A confocal microscopy study” with “A confocal microscopy approach (or analysis)…”

-Line 511 change inmunolabeled with immunolabeled

-Line 516 have you got an idea about how many JE22-NPs are located on the cell surface of CD44+ tumor cells and interact with on?

I can conclude that this manuscript is suitable for publication in Nanomedicine and Nanotechnology section of Pharmaceutics Journal, after some other experiments and minor recommended revisions.

Reviewer 3 Report

This article entitled "Selective anticancer therapy based on CD44 inhibitors loaded 2 nanoparticles" by Jose M. Espejo-Román et al. reported a new development of CD44 inhibitor conjugated polymeric nanoparticles via hydrazone bond to render a significant therapeutic effect. Their findings are very interesting and important because the pH sensitive strategy to release the CD44 inhibitor from the nanodevice at the acidic tumor microenviroment enhance the efficient interaction with CD44 receptor and significantly reduce HA binding. I recommend some minor modification.

  1. All the figures and text are small and blurry especially Figure 3. It would be better to make the text bigger and increase the resolution.

  1. In Figure 3 or 4, it is better to show the DLS data in acidic condition like pH5. The size differences can be shown in order to show the change before and after cleavage due to pH change.

  1. In Figure 5 and S9, in order to assess the biological activity of this nanodevice to target CD44, CD44-negative MCF-7 cells can be used as a negative control.

  1. In Paragraph 3.3 and Figure 6 and 7, the title is “3.3. Evaluation of efficiency of the designed nanodevice for the inhibition of CD44 receptor”. It looks not clear. It seems better to use the expression “inhibition of CD44 receptor binding” rather than “inhibition of CD44 receptor”. To show more biological mechanism, it is better to show RT-qPCR or Western Blot analysis data to prove the inhibition of CD44 expression.

  1. In Figure 8, in addition to Scratch-wound migration assay, it can be confirmed through other experiments like transwell migration assay, transwell invasion assay or 3D spheroid assay to show that CD44 proteins can stimulate tumor cell proliferation, motility, and invasion.

  1. It will be better to show the efficacy of the drug delivery system in animal tumor models.

  1. There are many typo errors like “significatly” (page 16 line 610). A word “Computacional” looks like a Spanish. Please change to “Computational”. Please find and correct all typo erroes in the article.

Round 2

Reviewer 3 Report

The authors selectively responded some of the concerns and have not addressed some of the major concerns.